# Enhanced Gas Separation Prowess Using Functionalized Lignin-Free Lignocellulosic Biomass/Polysulfone Composite Membranes

**DOI:** 10.3390/membranes11030202

**Published:** 2021-03-13

**Authors:** Abiodun Abdulhameed Amusa, Abdul Latif Ahmad, Adewole Kayode Jimoh

**Affiliations:** 1School of Chemical Engineering, Engineering Campus, Universiti Sains Malaysia, Nibong Tebal 14300, Pulau Pinang, Malaysia; aabiodun581@gmail.com; 2Process Engineering Department, International Maritime College, Sohar 322, Oman; adewolejk@gmail.com

**Keywords:** composite membranes, carbon dioxide, lignocellulosic biomass, pretreatment, amine groups

## Abstract

Delignified lignocellulosic biomass was functionalized with amine groups. Then, the pretreated lignin-free date pits cellulose and the amine-functionalized-date pits cellulose (0–5 wt%) were incorporated into a polysulfone polymer matrix to fabricate composite membranes. The amine groups give additional hydrogen bonding to those existing from the hydroxyl groups in the date pits cellulose. The approach gives an efficient avenue to enhance the CO_2_ molecules’ transport pathways through the membrane matrix. The interactions between phases were investigated via Fourier transformed infrared spectroscopy (FTIR) and scanning electron microscopy (SEM), whereas pure gases (CO_2_ and N_2_) were used to evaluate the gas separation performances. Additionally, the thermal and mechanical properties of the fabricated composites were tested. The pure polysulfone membrane achieved an optimum separation performance at 4 Bar. The optimum separation performance for the composite membranes is achieved at 2 wt%. About 32% and 33% increments of the ideal CO_2_/N_2_ selectivity is achieved for the lignin-free date pits cellulose composite membrane and the amine-functionalized-date pits cellulose composite membrane, respectively.

## 1. Introduction

The manufacturing industries capitalize on industrial gases. These gases are supplied in different categories ranging from pipeline down to supplies in gas cylinders. The public and the tradesmen hire associated equipment and gas cylinders for domestic and commercial purposes. Additionally, medical oxygen, welding gases, and balloon helium are other aspects covered by this product [1,2,3]. Thus, unwanted gases such as carbon dioxide (CO_2_) are released into the atmosphere as a result. The rise in environmental air pollution is attributed to inappropriate “from the source” capture of the greenhouse gases. The researchers from different fields in the scientific community are greatly concerned about the increasing discharge of CO_2_ into the atmosphere which is resulting in the phenomenon of climate change catastrophe [4]. The industrial gases cornerstone purification technologies such as hydrogen reforming and air separation require a lot of gas separation schemes and steps which are tedious [5]. Membrane technology is considered as an alternative. Industrial and laboratory scale-up have adopted polymeric membranes because it saves energy, environmental-friendly, and high product quality [6,7,8].

As of today, the investigated polymeric membranes are short-of withstanding dimensional, thermal stability, mechanical stability, and durability with expected excellent perm-selectivity under high processing conditions. The drawback of gas separation performance is attributed to the permeability and selectivity trade-off called the upper bound limit [9]. The polymer permeability to gases is affected by their low packing efficiency and low density. The permeability controls the movements of the permeating molecules through the membranes. Therefore, composite (mixed matrix) membranes (CMBs) filled some of those loopholes [10,11]. 

CMBs entail incorporating organic or inorganic materials (fillers) into the polymer matrix to explore the available micropores or functional groups for better interaction and efficient gas separation [12,13]. In comparison to the pure polymeric membranes, CMBs have been reported to show improved selectivity and higher permeability [14]. For the economical purpose, optimum and efficient separation can be achieved by maximizing the permeability and selectivity of the membrane. The multiplication of diffusivity and solubility coefficients results in permeability. The diffusivity and solubility can be improved by incorporating inorganic materials that possess the features of molecular sieve and using polymers that contain functional groups, e.g., oxygen-ether, amine, and hydroxyl, respectively. One of the natural (organic) fillers that can be used as molecular sieves is lignocellulosic biomass (LCB). The functionalization of the LCB with any of these functional groups can assist in improving the solubility [15]. 

LCB consists of polyols components that can be capitalized on. This is achievable when production optimization and quality improvement of LCB is assured. Recalcitrance effects of lignin can be prevented by adopting adequate pretreatment techniques that ease accessibilities to cellulose and hemicellulose in the LCB [10]. This will also increase the surface area to volume ratio with less weight, stiffness, and higher strength. Numerous industrial applications can benefit from any bio-composites developed which can be an excellent substitute for the available reinforcing agents and contactor systems [16]. 

In this study, the aim is to investigate the gas performance of CMBs incorporated with pretreated and functionalized lignin-free LCB filler on the selectivity performance and the CO_2_ and N_2_ permeability of the CMBs. LCB was collected from consumers, pretreated, functionalized, and used for the fabrication of different CMBs by being incorporated into the polysulfone (PSF) polymer matrix. PSF was chosen as a polymer for this study due to its thermal and mechanical stability which resulted in good membrane performances like selectivity and gas permeability. Different analytical techniques were used to characterize the fabricated CMBs and then gas (CO_2_ and N_2_) separation and permeation tests followed.

## 2. Membranes Mechanisms 

### 2.1. Materials and Chemicals Used

Date pits (Sukkari, main LCB) from Riyadh, Saudi Arabia. Epichlorohydrin (EPH, 99%), ammonium hydroxide (28–30% NH_3_), dimethylsulfoxide (DMSO, 99.9%), tetrabutylammonium hydroxide (TBAH, 98%), 1-methyl-2-pyrrolidone (NMP, 99.5%), and tetrahydrofuran (THF, 99.5%) were used without any further purification as purchased from Sigma Aldrich. The polysulfone (PSF beads) was purchased from Amoco Chemicals. The deionized (D.I.) water was used as provided. 

### 2.2. Lignin-Free LCB Functionalization

The delignified date pits cellulose (DPC) was obtained by adopting the method reported elsewhere [17]. Quantifying the efficiency of delignification is beyond the scope of this study. The functionalization was carried out using the method reported by Akhlaghi et al. [18] with some modifications. An amount of 1.00 g of the lignin-free DPC was dispersed in a round-bottomed flask containing the mixture of 66.66 mL DMSO and 1.00 g of TBAH. A syringe was used to drop the formed solution drop-wisely from another reflux set-up containing 3.78 mL of ammonium hydroxide and 1.46 mL of EPH that was heated for 2 h at 65 °C. Then, the reaction mixture under reflux was heated at 65 °C and stirred for 5 h. The reaction mixture was separated by centrifugation at 40,000 rpm for 5 min and washed three times using D.I. water before being free-dried. The dried amine-functionalized lignin-free LCB (DPC-NH_2_) was kept in a glass bottle for further analysis. 

### 2.3. Fabrication of Pure PSF and Lignin-Free Composite Membranes

Firstly, the pure PSF and lignin-free composite membranes were used as the top layer of the membranes. THF solvent was used to dissolve an 18 wt% oven-dried PSF to fabricate pure PSF membrane at 60 °C on a hot plate with stirring for 24 h using the modified mixing solution casting method reported in Ismail and Lai [19]. Degassing to remove trapped bubbles followed for 4 h. The resulting dope solution was poured on a glass plate and cast using an appropriate specific clearance with a drawknife. Then, for Lignin-free composite membranes, the submicron-sized DPC and DPC-NH_2_ were used as fillers added to the pure PSF solution to form the composite membranes. Fillers (0–5 wt%) were incorporated into a pure PSF dope solution to fabricate CMBs with stirring on the hot plate. Degassing and casting followed like the pure PSF. The membranes were dried for 48 h in the open air. Then, the casting solution for the membrane support was prepared on a glass plate consisting of PSF, glycerol, and NMP as 25 wt%, 10.7 wt%, and 64.3 wt%, respectively. Solvent evaporation was expected while allowing a free-standing time of 30 s because others (10 and 60 s) did not give the desired morphology, before immersing into water containing coagulation bath for 24 h at room temperature and then into methanol for 2 h to achieve complete phase separation process. The membranes were undressed from the glass plate and dried for 48 h in the open air. Finally, the microporous support was laminated with the dense top layer while the characterization and other studies followed. The fabricated membranes were labeled PSF-DPC-0/PSF-DPC-NH_2_-0 indicating the pure PSF membrane. The CMBs were labeled PSF-DPC-1/PSF-DPC-NH_2_-1, PSF-DPC-2/PSF-DPC-NH_2_-2, PSF-DPC-3/PSF-DPC-NH_2_-3, and PSF-DPC-5/PSF-DPC-NH_2_-5 to indicate the incorporation of 1, 2, 3, and 5 wt% of DPC or DPC- NH_2_, respectively. 

### 2.4. Characterizations of Materials

CHNS Elemental Analyzer (Perkin Elmer 2400 Series II) at combustion temperature of 925 °C was used to carry out the elemental compositions of the samples (DPC and DPC-NH2). The difference was used to calculate the amount of oxygen in the samples. The solubilized lignin content from the extractive free lignocellulosic biomass was quantified using the acetyl bromide method [20] and followed with the determination of the Kappa number of the samples via the ISO 302:2015 standard. The methods reported by Mansor et al. [21] with some modifications were used to evaluate the amount of cellulose, hemicellulose, and lignin in the samples. The morphology of the DPC and DPC-NH_2_ fillers and the fabricated membranes was examined using scanning electron microscope SEM FEI (Quanta 450 FEG). The fabricated membrane samples were carefully broken and soaked in liquid nitrogen to ease fracturing and ensuring the microstructure was preserved and coated with a thin gold layer to examine the surface and cross-sectional membrane morphology. The membranes were also subjected to FTIR analysis using (IRAffinity-1S spectrophotometer Shimadzu). 

### 2.5. Gas Separation Performance Evaluation

The test system for the membrane gas separation and permeation evaluation had a membrane gas cell that housed the membrane connected to it. Then, the membrane gas separation and permeation system were operated at 35 °C and were separately fed with 99.99% pure CO_2_ or N_2_ gas. A bubble flow meter was used to measure the permeate flow. The range between 1 to 5 bar pressure difference was used. The CO_2_ or N_2_ gas permeance, ${\left(\frac{P}{l}\right)}_{i}$, was calculated by using Eq 1 and the triplicate of each gas permeation measurement was ensured. The permeance adopted and the prevalent unit is GPU (whereas 1 GPU = $10^{-6}\;{\mathrm{cm}}^{3}\left(STP\right)/{\mathrm{cm}}^{2}\;s\;\mathrm{cmHg}$).
(1)$${\left(P/l\right)}_{i}=22,414\;Fp_{1}{\left(ART\Delta p\right)}^{-1}$$
where *i* = CO_2_ or N_2_, *P* is the permeability, *l* is the membrane thickness, *F* (${\mathrm{cm}}^{3}/s$) is the permeate flow, *A* (11.341 cm^2^) is the membrane active permeation area, R (6236.56 ${\mathrm{cm}}^{3}\mathrm{cmHg}/\mathrm{molK}$) is the universal gas constant, T (K) is the absolute temperature, $\Delta p=\left(p_{2}-p_{1}\right)$ is the pressure change for component i, *p*_2_ and *p*_1_ are the feed and the permeate pressures, respectively.

The membrane ideal selectivity, *α*, was calculated by using Equation (2).
(2)$$\alpha =P_{{\mathrm{CO}}_{2}}\;{\left(P_{N_{2}}\right)}^{-1}$$

## 3. Results and Discussion

### 3.1. Morphological and Thermal Investigations

The mechanically pretreated LCB resulted in powdered particles. The particle size of the lignin-free powder which was pretreated was in our earlier work [17]. The Zeta Potential Analyzer (Zeta Nano Z) via the dynamic light scattering technique was used to study the particles’ dispersity of the samples. In general, dispersity is a dimensionless quantity that is measured between 0 and 1. The obtained heterogeneity of the DPC and DPC-NH_2_ is 0.142 ± 0.001 and 0.140 ± 0.006, respectively. The acceptable monodispersed particle size distribution is attributed to polydispersity index values that are less than 0.25 [22]. Figure 1 shows the SEM image of the DPC and DPC-NH_2_ at 3 µm. The micrograph image of Figure 1 showed that the processing and pretreatments could have resulted in the agglomeration of some of the large particles. Future research work would be extended to cover the prevention of agglomeration and other related issues such as the particles’ porous properties and the schematic diagrams for the surface modifications steps. However, pores are formed on the surfaces of DPC and DPC-NH_2_ as shown in Figure 1a,b, respectively. The porous structures could be attributed to the pretreatments. The DPC had a greater pore area than the DPC-NH_2_. The reason for the low pore area of DPC-NH_2_ could be due to the further pretreatment involving the presence of an amine group after the functionalization of DPC. Other reagents facile access would be easy via the mesopores presence after the pretreatments and functionalization. The composition of these fillers will be investigated in the subsequent section.

Figure 2 and Figure 3 show the surface SEM micrographs of the dense top layer at 20 µm, lignin-free composite membranes, with different loadings of DPC and DPC-NH_2_, respectively. The morphologies of the composite membranes were discussed in comparison to the pure PSF membrane. Figure 2a and Figure 3a represent the surface SEM micrograph of the pure PSF which is smooth, defect-free, and no detectable micro pores exist. Successful incorporation of DPC and DPC-NH_2_ into the PSF is indicated by the observed particles and all the surfaces are free of potholes. The DPC and DPC-NH_2_ incorporation up to 2 wt% produced composite membranes that are void-free. However, agglomeration exits for other fillers loading into the polymer matrices as observed in Figure 2 and Figure 3d,e. Figure 4a (at 50 µm) shows that the membrane support has a porous top layer. The cross-sectional SEM micrograph of the assembled composite membrane at 20 µm also confirmed the porous nature of the support as shown in Figure 4b. The assembled composite shows that the top layer had a distinct morphology different than the porous support. The pores of the support are uniformly distributed in a spongy-like form. Likewise, the top layer and the support interface reveals good lamination. The later section will elaborate on the compositions of these materials.

### 3.2. Composition and IR-Spectra Analyses

Table 1 shows the composition of DPC and DPC-NH_2_. The DPC-NH_2_ sample contained atomic nitrogen as revealed from the elemental analysis. The results are within the value range for LCB reported elsewhere [23,24]. The absence of nitrogen in DPC confirms that the surface modification of DPC with amine groups was a success [25,26]. The sulfate ester group hydrolysis via pretreatments could be responsible for the decrease in the sulfur content in DPC-NH_2_ compared to DPC [27]. The evaluated nitrogen content from the elemental analysis was substituted into Equation (3) to determine the degree of surface substitution (DSS) [28]. The calculated DSS value is 0.42 which indicates that a higher amine content resulted using the protocol of amine functionalization reported in this study compared to 0.02 [29], 0.03 [30], 0.08 [31], and 0.32 [32] that were found in the literature. This could be attributed to the delignification of the LCB which reduces the resistance to other constituents in biomass. Three tests were carried out and the average values of the Kappa numbers for DPC and DPC-NH_2_ are presented in Table 1. The decrease in the Kappa number indicates the delignification efficiency because a direct relationship exists between the lignin amount in a sample and the Kappa number. This conforms to the increase in the amount of holocellulose (total cellulose and hemicellulose) for DPC, and DPC-NH_2_ as 96.88, and 97.13 wt%, respectively [33]. As presented in Table 1, the changes in the 2 θ values of the samples indicate the disruption of some portions of the methyl and methylene in the samples. More so, the lattice parameters, e.g., tensile strain, changed resulting in the shift of the peaks to higher diffraction angles and d-spacing. These observations are attributed to the successful delignification and amine functionalization of the LCB [34,35]. Therefore, the inherent functional groups were investigated subsequently via spectra study.
(3)$$DSS=\left(162\left(N\%\right)\right){\left[1400-108.56\;\left(N\%\right)\right]}^{-1}$$

The Fourier Transform Infrared (IR-spectra) of DPC and DPC-NH_2_ is shown in Figure 5. The cellulose and hemicellulose components of DPC and DPC-NH_2_ have contributions to the absorption band at 2870 cm^−1^ which is attributed to the C-H stretching absorption bands and the characteristic broadband at 3200–3400 cm^−1^ indicates the O−H stretching vibration bands [36]. The new peak at 1140 cm^−1^ is attributed to an amide I. This characteristic band at 2870 cm^−1^ indicates the appearance of N−H bonds in DPC-NH_2_ and hence proves the successful amine-functionalization of DPC. Additionally, the N-H primary amine band is attributed to the two bands on the broad O-H band region at 3360 and 3470 cm^−1^ [37]. These peaks are indicating a successful functionalization of amine groups on the surface of the DPC.

Figure 6 shows the FTIR spectra of pure PSF membrane and CMBs incorporated with DPC and DPC-NH_2_. As indicated in Figure 6a–c), the C–H stretching is attributed to the band at 2960 cm^−1^ and the bands between 1580 and 660 cm^−1^ are attributed to the O=S=O stretching vibrations. The IR-spectra of the CMBs showed that the incorporation of DPC and DPC-NH_2_ retained the characteristic bands of the PSF. The broadband at 3360 cm^−1^ is attributed to the presence of an OH group on the CMBs, which was missing on the IR-spectra of the pure PSF membrane. This tandem with the IR-spectra study reported by Pouresmaeel-Selakjani et al. [38]. These indicate the presence of DPC and DPCNH_2_ in the CMBs.

### 3.3. Gas Performance Measurement

The CO_2_ permeance and the ideal CO_2_/N_2_ selectivity for the coupled pure PSF membrane were studied for pressure difference between 1 to 5 bar to identify the optimum pressure. As presented in Figure 7, the optimum pressure is 4 bar. As pressure increases, the CO_2_ permeance decreases while the N_2_ permeance almost remains constant. For situations like this, the common behavior of glassy polymers is that the Langmuir sorption sites are being contested by the gas molecules [39]. Later, Henry’s law of simple dissolution transport takes control which resulted in weaker contributions from the Langmuir region. Afterward, a constant permeance value will be approached [40,41]. Thus, the CMBs were studied at 4 bars to investigate the effects of the fillers (DPC and DPC-NH_2_). The CO_2_ permeance and N_2_ permeance of the CMBs incorporated with different filler loadings are presented in Table 2. Increasing the filler loading increases the CO_2_ permeance and N_2_ permeance. This could be attributed to the surface modifications that increase access to the OH-groups in the fillers which increases the surface reactivity of the CMBs [42,43]. Figure 8 shows the ideal CO_2_/N_2_ selectivity of the CMBs incorporated with different filler loadings. The ideal CO_2_/N_2_ selectivity increased due to an increase in the CMBs’ affinity toward CO_2_ via the OH and the amine group’s presence. The observation indicates that the mesoporosity of DPC and DPC-NH_2_ were not blocked by the polymer chains penetration and the high molecular dimensions difference between the CO_2_ and N_2_ molecules [44]. The 2 wt% DPC and DPC-NH_2_ incorporated in the CMBs resulted in about 32% and 33% increments of the ideal CO_2_/N_2_ selectivity, respectively. Thus, the optimum filler incorporated CMBs used in this study in comparison to the optimum coupled pristine PSF membrane showed an enhanced performance. Nonetheless, fillers loading increment from 2 to 5 wt% resulted in the ideal CO_2_/N_2_ selectivity decrease, as presented in Figure 8. This observation might be attributed to the agglomeration of the fillers at higher loading in the CMBs [45]. Therefore, the need to investigate the mechanical and thermal properties of the optimum membranes becomes inevitable.

### 3.4. Effect of Temperature on The Optimum Membranes

The optimum membranes’ permeance for CO_2_ and N_2_ gases were investigated as a function temperature in the temperature range of 25–75 °C. The outcome of the test as shown in Figure 9a,b conforms with the gas transport trends for glassy polymers for the CO_2_ permeance and N_2_ permeance, respectively [46]. The CO_2_ permeance increases as the temperature increases which could be attributed to the increase in flexibility within the polymer matrix. However, an increase in temperature causes the N_2_ permeance to increase more. Thus, the ideal CO_2_/N_2_ selectivity decreases as temperature increases. This phenomenon could be described using the gas permeance relationship in the Arrhenius equation (Equation (4)) with the operating temperature [47]. The behavior of the gas permeance as time increases, called the aging effect, is covered in the scope of our future study. This will show the effect of the membrane’s non-equilibrium state as it affects the free volume whether the membrane becomes thinner with time. Additionally, the external factors, e.g., moisture, contamination effects will be investigated.
(4)PP0=exp−ΔEpRT
where $P_{0}$ (permeance) is the permeability of the gas pair, and $\Delta E_{p}\;\left(\mathrm{kJ}/\mathrm{mol}\right)$ the activation energy of the material, T (K) is the temperature, R ((kJ·mol^−1^)·K^−1^·mol^−1^) is the gas constant.

### 3.5. Mechanical and Thermal Properties of the Optimum Membranes

The Gotech Testing Machine (GT-7010-D2) was used based on the standard method (ASTM D412) to measure the membranes’ tensile strength at 50 mm clamp distance with a crosshead speed of 20 mm/min. The mechanical properties of filler incorporated composites increase via the fillers and the polymer composite adhesion as presented in Table 3. The small size of the fillers and their strong surface activities enhance the capacities of the polymers to a certain level. Thereby enabling the OH groups to increase the PSF and fillers interaction via increased hydrogen bonding and molecular force. As such, the positive effects of the incorporation of the filler also resulted in the optimum fabricated CMBs possessing an increased tensile strength. The obtained values for cellulose-based fillers incorporated into PSF are within the accepted range reported by Bai et al. [48]. The Perkin–Elmer (DMA 7e) dynamic mechanical analyzer damped at 100 mN/min from 100 to 4000 mN was used to measure Young’s modulus of the membranes. The incorporation of fillers into the polymer matrix to form CMBs caused an increase in Young’s modulus from 2.02 ± 0.04 to 2.63 ± 0.02 GPa. This observation can be attributed to contributions from the well-dispersed non-agglomerated fillers used and the polymeric chains experiencing good surface adhesion with the fillers. The long-life performance of these CMBs would be good because of the strong interaction between the fillers and the polymers via multiple hydrogen bonding formations [49]. The tensile strength and Young’s modulus simultaneous increment can be attributed to the delay of cracks because the polymer segments experiencing the coupling behavior of the fillers via physical transfusion [50]. Furthermore, model DSC 7 of differential scanning calorimetry from Perkin-Elmer was operated to measure the membranes’ glass transition temperature (T_g_) from 50 to 200 °C and at 10 °C/min of temperature and heating rate, respectively. The declining T_g_ values can be attributed to the plasticizing effect due to polymer chain disruption that resulted in the formation of dense polymer package [51]. However, Young’s modulus and the tensile strength increment prove that other justifications will be required to confirm the plasticizing effect. Thermogravimetric analysis (TGA) of the membranes was investigated after drying up to 500 °C from room temperature. Perkin Elma analyzer (TGA 7) was used in an inert atmosphere under the conditions of 20 mL/min, 10 °C/min, and 10 mg for the flow rate, heating rate, and weight of membranes samples, respectively. As shown in Figure 10, the degradation profile of the materials is similar such that differences can be observed in comparison to the pristine. The successful DPC and DPC-NH_2_ incorporations and strong adhesion are further established.

### 3.6. Results Comparison with Literature

The choice of PSF is to capitalize on its high chemical and mechanical stability and it shows better compatibility with biomaterials than other polymers [52]. Additionally, fine particles (<20 µm) of bio-fillers can withstand degradation because the mechanism at the size is complex [53]. Thus, the stability of the fabricated CMBs in this study could be guaranteed. However, the upper bound (Figure 11) is not surpassed in this study considering the gas separation performance and gas permeation of the fabricated CMBs incorporated with DPC and DPC-NH_2_ [54]. Consequently, further studies are necessary on pretreated and surface modified lignocellulosic biomass composite membranes to improve the gas separation performance and permeation. The cellulose incorporated CMBs from the literature comparison with the CO_2_ permeance and the ideal CO_2_/N_2_ selectivity in the current study presented in Table 4. Several reported studies on CO_2_ permeance are comparable with the obtained CO_2_ permeance in the current study. As shown in Table 4, the incorporation of 2 wt% of DPC and DPC-NH_2_ filler in the current study achieved a relatively higher ideal CO_2_/N_2_ selectivity.

## 4. Conclusions

Fabrication of composite membranes by incorporating pretreated and functionalized lignin-free lignocellulosic biomass into PSF membrane matrix in a simple method was presented. The amine functionalization of the lignin-free DPC and successful incorporation into CMBs as observed in the FTIR, SEM, and TG. Additionally, separation performance was significantly enhanced. Under optimum conditions, the tests confirmed that fillers (DPC and DPC-NH_2_) incorporated CMBs displayed an enhanced and better performance than the pristine PSF membrane. The increased permeance resulted via lignin-recalcitrance removal and capitalizing on the OH-group. The added amine groups further increased the mesoporous structure of the PSF-DPC-NH_2_ than the PSF-DPC. It can be stated, concerning the data from the literature, that cellulose and amine-functionalized cellulose from the agricultural wastes can be used as fillers for scalable technology. Further studies are required to obtain concrete results on the investigation of cellulose-based inside-membrane gas transport mechanism to achieve high CO_2_ permeance and high selectivity. The cost evaluation of CMBs fabrication for industrial application possibilities should be considered to expand the scope of lignocellulosic biomass valorization.

## Figures and Tables

**Figure 1 membranes-11-00202-f001:**
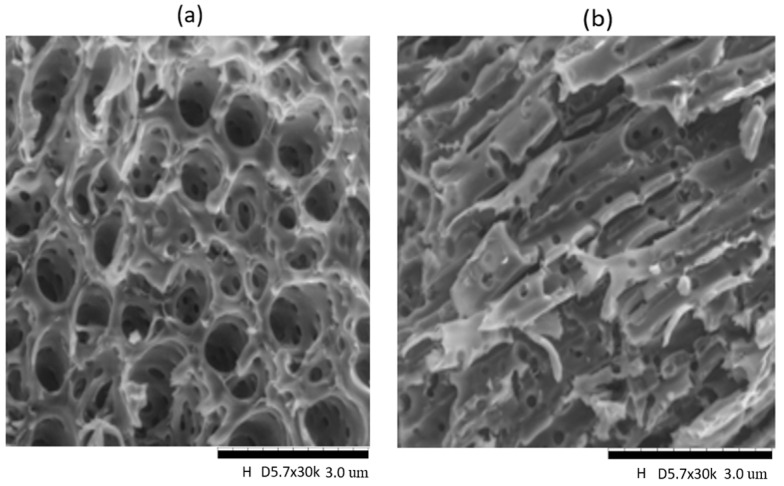
SEM image of (**a**) date pits cellulose (DPC) and (**b**) dried amine-functionalized lignin-free LCB (DPC-NH_2_).

**Figure 2 membranes-11-00202-f002:**
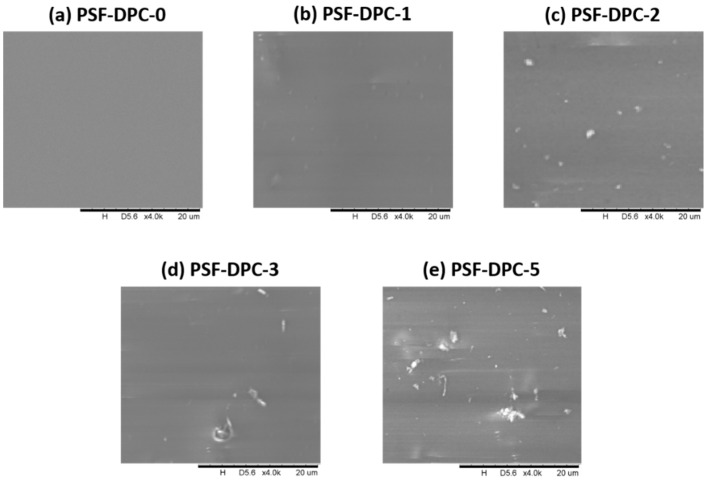
Surface SEM of DPC composite membrane for (**a**) PSF-DPC-0, (**b**) PSF-DPC-1, (**c**) PSF-DPC-2, (**d**) PSF-DPC-3, and (**e**) PSF-DPC-5, for the pure PSF membrane and the CMB’s with the DPC loads of 1 wt%, 2 wt%, 3 wt%, and 5 wt%, respectively.

**Figure 3 membranes-11-00202-f003:**
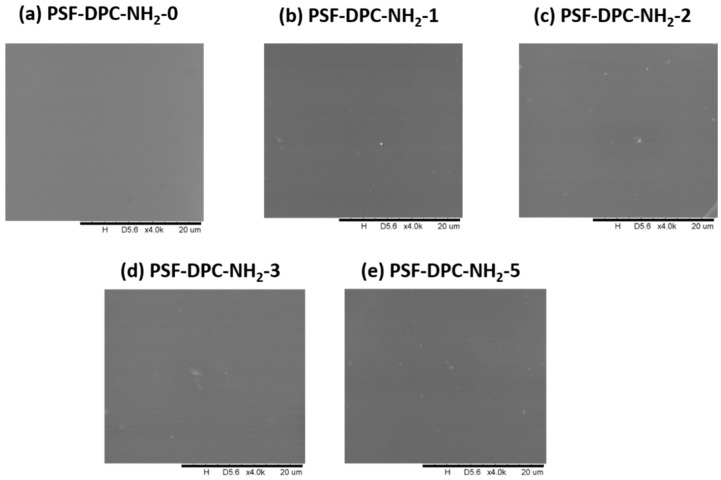
Surface SEM of DPC-NH_2_ composite membrane for (**a**) PSF-DPC-NH_2_-0, (**b**) PSF-DPC-NH_2_-1, (**c**) PSF-DPC-NH_2_-2, (**d**) PSF-DPC-NH_2_-3, and (**e**) PSF-DPC-NH_2_-5, for the pure PSF membrane and the CMB’s with the DPC loads of 1 wt%, 2 wt%, 3 wt%, and 5 wt%, respectively.

**Figure 4 membranes-11-00202-f004:**
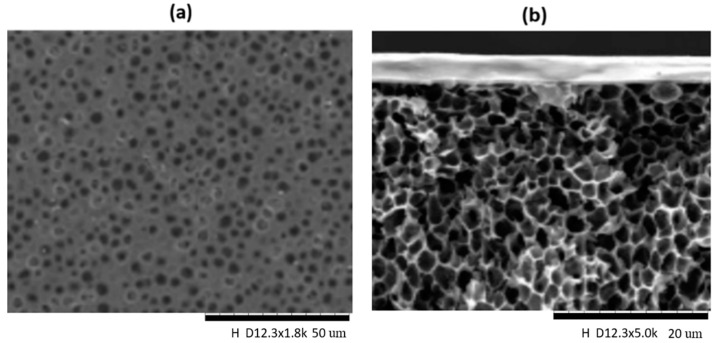
SEM image of (**a**) the top surface of the porous support and (**b**) the cross-section of dense top and porous support of composite membrane.

**Figure 5 membranes-11-00202-f005:**
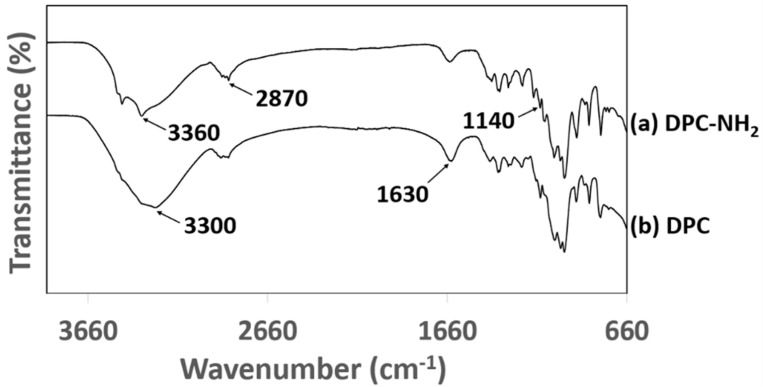
IR spectra of (**a**) DPC and (**b**) DPC-NH_2_.

**Figure 6 membranes-11-00202-f006:**
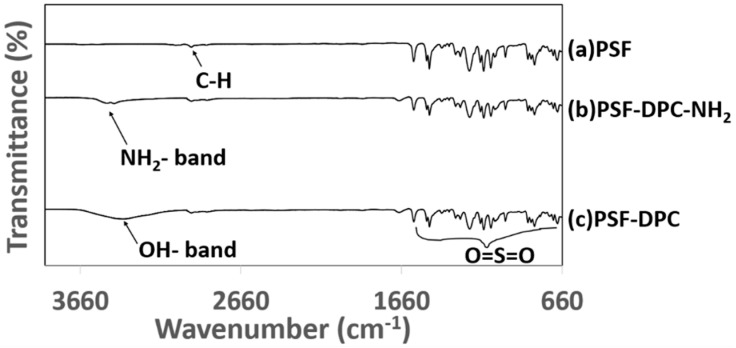
IR-spectra of membrane samples (**a**) polysulfone (PSF), (**b**) PSF-DPC-NH_2_, and (**c**) PSF-DPC.

**Figure 7 membranes-11-00202-f007:**
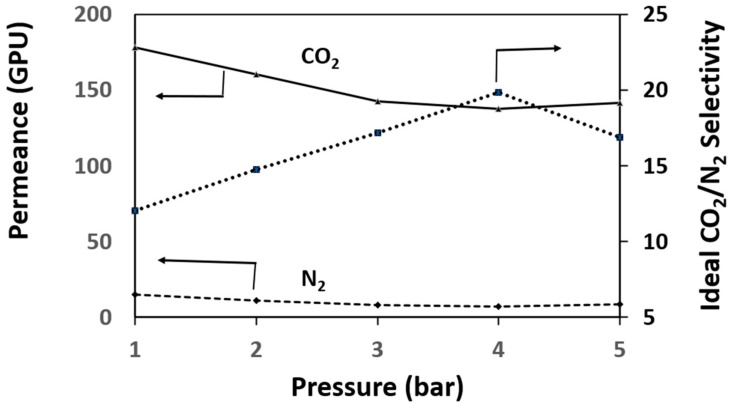
Optimum pressure for the coupled pristine PSF membrane.

**Figure 8 membranes-11-00202-f008:**
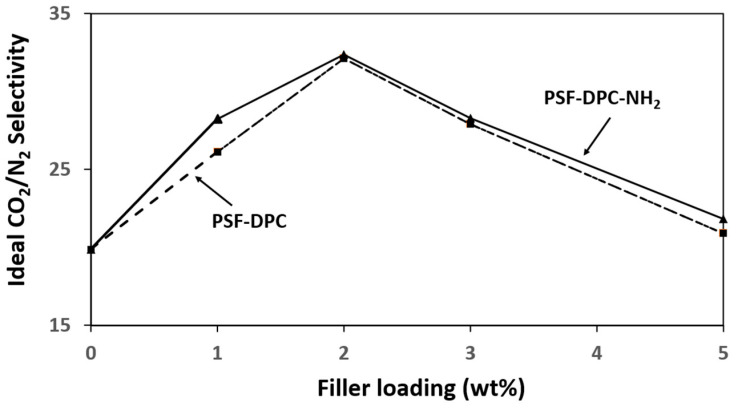
Ideal CO_2_/N_2_ selectivity of the CMBs incorporated with different DPC and DPC-NH_2_ loadings at a pressure difference of 4 bar.

**Figure 9 membranes-11-00202-f009:**
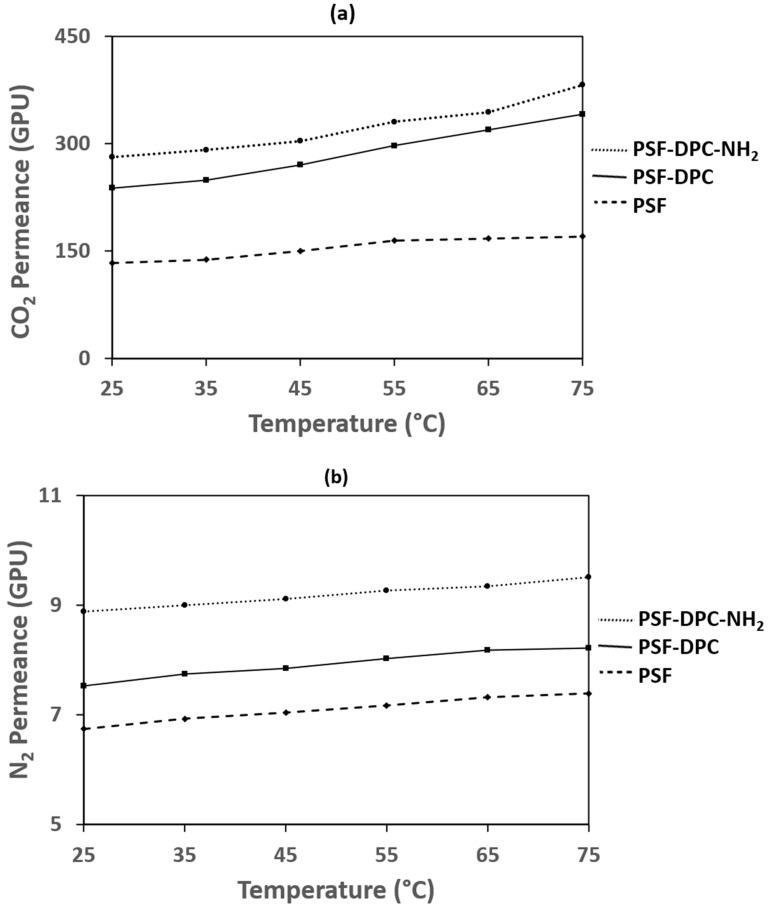
Effect of temperature on the optimum membranes’ (**a**) CO_2_ permeance and (**b**) N_2_ permeance.

**Figure 10 membranes-11-00202-f010:**
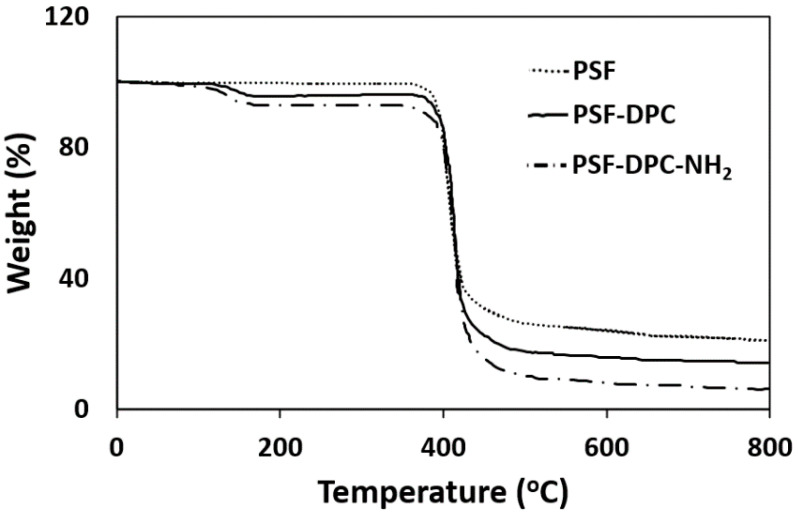
TGA thermogram of optimum coupled pristine PSF and composite PSF incorporated with DPC and DPC-NH_2_.

**Figure 11 membranes-11-00202-f011:**
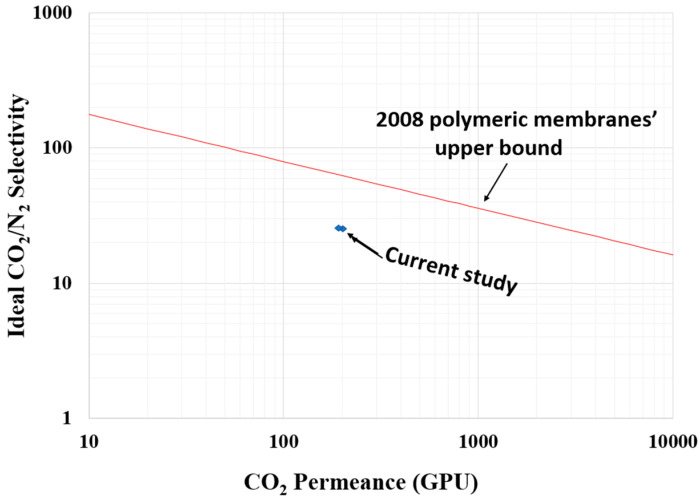
Comparison of study with the upper bound limit.

**Table 1 membranes-11-00202-t001:** Composition of DPC and DPC-NH_2_.

Samples	Carbon, C (%)	Hydrogen, H (%)	Nitrogen, N (%)	Sulfur, S (%)	Hemicellulose (wt%)	Cellulose (wt%)	Kappa Number	XRD	
								2 θ (°)	d-Spacing (Å)
DPC	54.01 ± 1.11	10.65 ± 0.04	0	0.79 ± 0.03	17.15 ± 1.40	79.73 ± 1.22	2.80	22.69	2.4
DPC-NH_2_	55.03 ± 0.78	11.31 ± 0.25	2.84 ± 0.07	0.75 ± 0.01	17.21 ± 1.01	79.92 ± 1.05	2.79	23.87	2.5

**Table 2 membranes-11-00202-t002:** CO_2_ gas permeance and N_2_ gas permeance of the composite (mixed matrix) membranes (CMBs) incorporated with different DPC and DPC-NH_2_ loadings at a pressure difference of 4 bar.

Filler Loading (wt%)	CO_2_ Gas Permeability (GPU)		N_2_ Gas Permeance (GPU)	
	PSF-DPC	PSF-DPC-NH_2_	PSF-DPC	PSF-DPC-NH_2_
0	137.65 ± 0.01	137.66 ± 0.03	6.93 ± 0.04	6.93 ± 0.05
1	164.13 ± 0.02	222.37 ± 0.02	6.29 ± 0.03	7.87 ± 0.05
2	248.84 ± 0.01	291.20 ± 0.01	7.75 ± 0.02	9.00 ± 0.01
3	322.97 ± 0.01	344.15 ± 0.02	11.58 ± 0.04	12.18 ± 0.03
5	358.32 ± 0.02	381.21 ± 0.01	17.16 ± 0.01	17.47 ± 0.02

**Table 3 membranes-11-00202-t003:** Mechanical and thermal properties of the optimum membranes.

Membrane	Tensile Strength (MPa)	Young’s Modulus (GPa)	T_g_ (°C)
Pristine PSF	2.52 ± 10.16	2.02 ± 8.24	165.3 ± 1.07
PSF-DPC	2.90 ± 10.23	2.51 ± 6.41	164.1 ± 0.98
PSF-DPC-NH_2_	3.16 ± 20.41	2.63 ± 7.82	162.6 ± 0.62

**Table 4 membranes-11-00202-t004:** Comparison of CO_2_/N_2_ selectivity performance with literature data.

Polymer-Filler	Filler Loading (wt%)	Pressure (bar)	CO_2_ Permeance (GPU)	Ideal CO_2_/N_2_ Selectivity	Reference
PU/PEBA	60	10	~95.00 *	~30.00	[55]
PEBAX/PEI-ZIF-8	5	30	~13.00	~49.00	[56]
PSF-Nanosilica	2	~2	~30.90 *	~7.70	[57]
PSF/fMCM-41	20	10	~9.13 *	~32.97	[58]
PSF/ZTC	0.4	5	58.50	11.62	[59]
PES/CNT	1	1	~10.90 *	~3.07	[60]
PSF/DP	2	10	~8.46	~1.65	[12]
PSF/DPC	2	4	~322.97	~32.11	Current study
PSF/DPC-NH_2_	2	4	~344.15	~32.35	Current study

PU: Polyurethane, PEBA: Poly(ether-block-amide), Fmcm-41: Functionalized mesoporous silica material, ZTC: Zeolite templated carbon, PES: Polyethersulfone, CNT: Carbon nanotube, DP: Date pits, * Gas permeability (in Barrer) was reported instead of gas permeance (in GPU).

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
