# Peer review of "Enhanced Gas Separation Prowess Using Functionalized Lignin-Free Lignocellulosic Biomass/Polysulfone Composite Membranes"

_membranes, 2021, doi:10.3390/membranes11030202_

Round 1

Reviewer 1 Report

The manuscript reports the preparation of composite membranes based on PSF and delignified lignocellulosic biomass fillers (functionalized or not) with improved gas separation performance. The subject of the manuscript falls in a very active area of the gas separation membranes R&D. In this regard the impact of the work could be good. However, the manuscript have a number of drawbacks which must be improved to be considered for publication:

1) First of all the English of the manuscript are very poor and need to be significantly improved. Not only in terms of grammatical errors but also in terms of writting style. Especially in the introduction it is very difficult for the reader to follow up.

2) Although the manuscript should focus more in the gas separation performance of the synthesized membranes, it mainly focus in materials characterization and mainly in the characterization of the delignified lignocellulosic biomass fillers. The authors must add more results related to the gas separation assessment of the membranes. For example some graphs presenting the effect of temperature and pressure on the gas separation performance of the optimum composite membranes can be added. 

Other issues that need improvement or clarification are the following:

  • Table 1 give some general information about the material properties (e.g. viscosity, autoignition temperature, flash point, etc.) not relevant to the discussion followed. The reader can easily find this information if needed. If the Table does not serve a specific purpose it has to be removed.
  • Please explain in details what the DP sample is standing for.
  • The format in Table 2 is a bit confusing. It has to be improved.
  • Please propose an explanation for the behavior observed with pressure. Comment based on literature data about PSF membranes.
  • In Table 3 and Figure 8 please include data for 0% fillers concentration (pure PSF membranes).
  • The authors report that the gas separation performance of the synthesized membranes are below the upper bound but no data about the upper bound are presented. Please present the results in comparison with the upper bound, e.g. in a plot.
  • In Table 5 please add more references, focusing in PSF based membranes.

Author Response

The response is attached for your review. Thank you.

Reviewer 2 Report

The article entitled “Enhanced Gas Separation Prowess Using Functionalized Lignin-free Lignocellulosic Biomass/Polysulfone Composite Membranes” is an interesting paper. The author fabricated a delignified lignocellulosic biomass and further bonding it with amine groups. Then, the pretreated lignin-free date pits cellulose and the amine-functionalized-date pits cellulose (0-5 wt%) were incorporated into a polysulfone polymer matrix to fabricate composite membranes. Matrix membranes will improve the transport of carbon dioxide molecules. However, some contents are inappropriate and incomplete. The manuscript has been recommended for major revision based on following observations.

  1. The main mechanism of this paper is screened, and the pore size has a great impact on screening performance, so the author should make corresponding tests.
  2. The authors need to add XRD patterns to prove that the delignified lignocellulosic biomass were successfully functionalized with amine groups.

3.In Fig. 7, the author should have drawn the permeability curve of N2 at different pressures to provide more support for the ideal selectivity.

  1. The pressure has a great influence on the gas permeation of the matrix membrane, and the author should make a corresponding explanation.
  2. Polymer membrane is easy to be age in practical applications, so the authors need to test the durability of synthetic matrix membrane in gas separation.
  3. There are many grammatical and spelling mistakes in the author’s writing. Please check the writing carefully and lots of grammatical and spelling mistakes were caused by halfheartedness.

Author Response

(The authors gave the same response as above.)

Reviewer 3 Report

The reviewer has read the manuscript entitled “Enhanced gas separation prowess using functionalized lignin-free lignocellulosic biomass/polysulfone composite membranes” carefully. The authors have reported incorporation of DPC and DPC-NH2 particles into PSF membranes to improve the separation performance. The reviewer thinks that this manuscript can be considered for publication after addressing below issues.

Comments

Some statements in the manuscript are problematic, e. g., “The molecule's permeability controls the movements of the permeating molecules through the membranes.” Permeability controls molecular movements.

If possible, please provide the chemical formulas or schematic diagrams of DP, DPC, and DPC-NH2.

“The obtained heterogeneity…. are 0.145 ± 0.006, 0.142 ± 0.001, and 0.140 ± 0.006, ….. less than 0.25”. What is the unit?

“Figure 1 shows the SEM image………observation  corroborates the high dispersity index of DP in comparison to DPC and DPC-NH2”. How did it prove? In fact, the images show bulk materials.

The images shown in Figure 2 and 4 seem to be compressed.

From SEM images, the particle sizes of DP, DPC, and DPC-NH2 seem to be larger than the dense layers of the membranes.

It is suggested to evaluate the porous properties of DP, DPC, and DPC-NH2.

Error bar reflect reproducibility of membranes. The error seems too small to be unrealistic.

Author Response

(The authors gave the same response as above.)

Round 2

Reviewer 1 Report

Authors made almost all the asked changes to improve the quality of the manuscript.

The only shortcoming of the manuscript, that needs to be improved, is the introduction. Although some improvements have already been made, it still seems to contain scattered, unconnected information. The wording must be improved and better coherence between the provided information is needed. 

Author Response

The introductory section has been improved as suggested. Thank you.

Reviewer 2 Report

The revised manuscript can be accepted for publication without further revision.

Author Response

Thank you.

Reviewer 3 Report

The manuscript can be published.

Author Response

Thank you.
